# Graph Machine Learning for Assembly Modeling

**Carola Lenzen**\*

Institute for Software & Systems Engineering, University of Augsburg, Augsburg, Germany
lenzen@isse.de

**Alexander Schiendorfer**\*

Institute AImotion Bavaria, Technische Hochschule Ingolstadt, Ingolstadt, Germany
Alexander.Schiendorfer@thi.de

**Wolfgang Reif**

Institute for Software & Systems Engineering, University of Augsburg, Augsburg, Germany
reif@isse.de

## Abstract

Assembly modeling refers to the design engineering process of composing assemblies (e.g., machines or machine components) from a common catalog of existing parts. There is a natural correspondence of assemblies to graphs which can be exploited for services based on graph machine learning such as part recommendation, clustering/taxonomy creation, or anomaly detection. However, this domain imposes particular challenges such as the treatment of unknown or new parts, ambiguously extracted edges, incomplete information about the design sequence, interaction with design engineers as users, to name a few. Along with open research questions, we present a novel data set.

## 1 Assembly Modeling

Assemblies are collections of parts that make up a product (see Figure 1). In computer-aided design (CAD), *assembly modeling* refers to designing a new product based on existing parts – think of a cabinet that consists of screws, doors, and hinges; or a bike that consists of a frame, wheels, etc [1]. The connection type (e.g., welding or fastening using bolts) may contain geometric information or constraints that are also part of the assembly model. By its very nature, assembly modeling gives rise to a number of interesting novel applications for graph machine learning. Note that assembly modeling in this paper refers to the act of (iteratively) *designing* a new product using the same library of existing parts whereas other lines of work emphasize the computer vision perspective of *perceiving* physical parts (e.g. [2]) or the 3D perspective of *constraining* pairs of parts according to their position and relative movement (e.g., [3]) – also using geometric deep learning. Our goal is to support design engineers, e.g., by suggesting next parts to insert or categorizing the existing parts by their usage.

Some challenges that manufacturing companies face are:

- Assemblies similar to existing ones frequently need to be designed and adjusted – in accordance to customer specifications (e.g., in special mechanical engineering).

- Knowledge about proven part combinations (e.g., particular hinges and doors, screws and bolts, . . . ) is available to senior design engineers and may follow a desirable part management but not made explicit and enforced in CAD software.

- Assembly models are produced in an *arbitrary sequence* which depends on the designer's individual preferences (e.g., start working on the front or back wheel of a bicycle is arbitrary); moreover, this insertion ordering is not stored in the final design by common CAD tools.

---

\*Equal contribution.

C. Lenzen et al., Graph Machine Learning for Assembly Modeling (Extended Abstract). Presented at the First Learning on Graphs Conference (LoG 2022), Virtual Event, December 9–12, 2022.

**Figure 1:** Assembly models (here, a jaw of a gripper) contain the structure of the included parts. Multiple instances of the same part type (here, A, B, C, D) may occur multiple times.

- Extracting a useful graph structure from CAD assembly models to begin with is not obvious. Although design engineers can define so-called "mates" relations between parts in a design to, e.g., define the rotation of a hinge, they are sometimes used for convenience in the CAD tool (cf. grouping elements) instead of actually denoting a physical connection or meaningful co-occurrence that could be reused.

In this extended abstract, we highlight opportunities for the graph machine learning community to work on CAD assembly modeling as a novel application along with an accompanying assembly data set [4]. Due to the symmetry properties in the data, graph ML is particularly suitable: as the CAD parts have no inherent order and their insertion sequence is not given (as it is not stored in final designs), permutation invariance is crucial e.g., for part recommendation.

Formally, in our problem setting we assume a set of part types $\mathcal{P}$ (e.g., a particular type of screw, hinge, etc., corresponding to physically existing parts) serving as vocabulary on which a data set of $N$ assemblies $\{A_i\}_{i=1}^N$ is based. An assembly $A_i$ specifies its containing parts (multiple instances of the same type are possible) as nodes $\mathcal{N}^{A_i}$, and information about connected parts as edges $\mathcal{E}^{A_i} \subseteq \mathcal{N}^{A_i} \times \mathcal{N}^{A_i}$. Each part $p \in \mathcal{N}^{A_i}$ is an instance of a part type, referred to as $\mathcal{T}(p) \in \mathcal{P}$. Consequently, each assembly is represented as an undirected, unweighted graph where the nodes are heterogeneous (different part types) and the edges are homogeneous (only one type of edges – expressing connectivity in a design – is allowed). This initial formulation is based on information that is always available while ignoring geometric information. Additional information (e.g., edge features) may be included in future work. Therefore, the parts can be represented by an one-hot encoding of their types or pretrained embeddings which may get extended in the future by additional features denoting geometric information, their material (steel, aluminium, or plastics), or properties of the parts such as conductivity or temperature resistance.

## 2 Graph ML Use Cases in Assembly Modeling

Given a data collection of assemblies as described, there are several application scenarios for graph machine learning. With respect to the distinction into structural (graph structure is explicit as in molecule generation) and non-structural scenarios (graphs are implicit and derived from text or images) given in [5], the proposed applications fall into a "semi-structural" category. The extraction of edges is a vital challenge of this application scenario as some CAD mates can be extracted canonically whereas others may have been defined by designers out of convenience with no actual meaning, so that some edges may need to be extracted from, e.g., geometric proximity. In all use cases, involving domain experts could be beneficial, for example by mitigating the side effects of ambiguous edges. However, we focused on mainly data-driven approaches due to the limited time design engineers devote to maintaining knowledge bases – even simple part taxonomies that are curated manually – which is why we consider this as a third use case.

### 2.1 Part Recommendation

In assembly modeling, design engineers can choose from a variety of existing part types, making selecting the right ones a cumbersome task. Past assemblies contain information both on the collection of used parts and on their combination to solve a specific task. We assume that parts which are used

together frequently are causally related and, therefore, parts that are likely to be inserted can be predicted using graph machine learning. In previous work [6], recommending next required parts during construction based on GNNs has already been investigated showing promising results – i.e., learning $P(\mathcal{P} \mid A_i^t)$ where $A_i^t$ refers to the state of an assembly at time step $t$. By contrast, the experiments revealed the inferiority of classical recommendation models like frequent itemset mining neglecting the graph structure. For the gripper jaw illustrated in Figure 1, next needed parts could be a handle and a screw to continue the construction of the entire gripper. Formulated as a graph classification problem where each class corresponds to a part type, the top-$k$ rate may be used as performance metric to rate multiple recommendations.

However, the presented approach does not incorporate *where* to assemble the recommended part to the partial design. This may be acceptable for small assemblies, but becomes unwieldy for large assemblies with plenty of possible extension nodes. A natural next step is to predict applicable parts for every already added part of the assembly – i.e., learning $P(\mathcal{P} \mid c \in A_i^t)$ to localize the part recommendation within an assembly. Due to this autoregressive nature of part recommendation, it bears similarity to graph generation although the focus is on incremental steps with user interaction as opposed to learning an assembly distribution $P(A)$ all at once, as will be discussed in Section 3. Since the main goal of part recommendation is to provide a design engineer with a small set of relevant parts to reduce the cognitive burden, approaches using conformal prediction [7] could prove useful.

## 2.2 Anomaly Detection of Mismatching Parts

A second (unsupervised) use case consists of detecting anomalies in assembly models such as an unexpected choice of particular part types (e.g., screws from a different manufacturer) or rarely used substructures that could hint at an unconventional way of solving a design task. From a business perspective, companies might limit their procurement to a set of well-known part types (better contracts with manufacturers, more reliable during the product life-cycle) within their strategic part management. Anomalous assembly models might emerge, e.g., from starting a new model based on a much earlier project with some part types having become obsolete or simply a lack of experience/knowledge on behalf of the design engineer. From a graph ML perspective, both identifying anomalous graphs in a database as well as identifying anomalous graph objects (nodes, edges such as, e.g., unexpected part types for the screws in the gripper in Figure 1) needs to be addressed [8], in particular to show to design engineers or procurers *where* and *how* the assembly is deviating.

The graph structure, i.e., the information about which parts are connected, also plays a central role in this use case: Considering the design of heavily loaded machines, so-called *predetermined breaking points* are often integrated, i.e. points that are supposed to break in case of overload for safety reasons. These components are only useful at certain locations of a construction. If the graph structure would be neglected and thus only the multiset of components would be considered, the information of the localization would be lost and an anomalous use of such components at wrong locations could not be detected.

## 2.3 Creating a Taxonomy of Parts

Third, using node embeddings $h_p$ of the parts $\{p \in A_i\}_{i=1}^N$ such as node2vec [9], DeepWalk [10], or comp2vec [6] for visualization and clustering could aid companies as well in their part management. The availability of well-curated, hierarchical taxonomies of part types depends on the level of maturity of a company and traditionally requires significant manual effort. A data-driven solution that exploits usage patterns in assembly models could organize a company's frequently used part types better. There has been an interest in making these embeddings (or latent representations) of nodes more interpretable to humans [11] which is what needs to be done for this task. However, the graphs retrieved from assemblies do not show homophily – two parts that are connected are most likely *not* similar but rather complementary (e.g., door and hinge) – which is an underlying assumption of many existing node embedding techniques. Here, synonymity in terms of usage (on which comp2vec is based) tends to be a better replacement, i.e., parts are similar if they can be used in the same contexts.

# 3 Related Work

The task of part recommendation bears some similarities to graph generation, i.e., approximating $P_{\text{data}}(G)$ with a parametrizable $P_{\text{model}}(G \mid \theta)$. The graph can be generated either all at once, for example using variational autoencoders[12] or generative adversarial networks[13], or incrementally by so-called *autoregressive* models that predict single or multiple nodes or edges step by step – conditioned on an intermediate state of the graph. Since our goal is to support design engineers by presenting suggestions instead of taking over the whole task, we go with the second approach. During CAD modeling, we want to allow changes on the partial assembly by designers. Therefore, we need a model that can generate arbitrarily large graphs which is typically not the case for non-incremental models.

Generating graphs with matching structural characteristics to the training data along with handling only one node type (i.e., $|\mathcal{P}| = 1$) as done using recurrent neural networks in GRAN [14] or GraphRNN [15] is not sufficient for our use case: the relevant information for predicting next needed parts lies both in the already used parts, i.e., the node features, and the structure of the graph. The focus is mainly on the type of part that should be added and only secondarily where to insert it into the existing graph. Since these approaches only evaluate the final graph structure (in particular, in terms of aggregated graph statistics such as degree distributions) without incorporating its intermediate states, canonical numbering of nodes can be performed for generating training instances, keeping their number small as no node permutations need to be considered. Common choices for GRAN or GraphRNN are breadth-first or depth-first traversals starting from the most connected node or random orderings. For assemblies, however, designers can start with any part or subgraph, followed by a generation sequence depending on the designer's preferences. Therefore, the authors in [6] create instances for every possible creation sequence of an assembly by iteratively cutting off nodes that serve as labels for the resulting partial assemblies. Unlike [16], in these approaches newly added nodes are always connected to the previous graph structure, which we want to enforce during construction.

Molecule graph generation refers to generating valid molecules with desired chemical properties, incorporating various types of nodes (i.e. different atoms) and even various types of connections (which is not necessary for using the current representation of assemblies). While guaranteeing the validity of the generated graph (like in [17] concerning the chemical structure of the generated molecule) may be assumed to be an important aspect for assembly modeling as well, this check-up turns out be not this obvious as the number of connection points of an part is typically not available or misleading since design engineers may adapt their geometry, e.g., by drilling holes, in order to assemble additional parts. However, this application domain seems to be the most similar to assembly modeling in terms of data representation. Nevertheless, again only the final generated graphs are relevant for evaluating the molecule generation model – as expounded above, also the intermediate steps matter for part recommendation.

# 4 Open Questions

The domain of assembly modeling imposes particular challenges that can stipulate further research in graph machine learning as described in detail in the following.

**How to deal with evolving data sets?** Over time, the part catalogs may get updated as well as new catalogs and part types may be incorporated to a company's part library. In particular at test time, we might be confronted with part types in assemblies that were not available during training. This setting confronts us with so-called *attribute-missing* graphs where all attributes of a subset of nodes are missing, opposed to *attribute-incomplete* graphs [18] that are composed of nodes all with non-empty attribute sets, typically treated by value imputation techniques either in a preprocessing step (e.g., [19] or [20]), or during processing the graph in the model (e.g., [21] or [22]). Methods based on the homophily assumption are not applicable since the assumption of connected nodes been similar is clearly violated in the assembly modeling use cases as connected parts typically serve different purposes. Initial work has been done on handling attribute-missing graphs, e.g., [18] that make a shared-latent space assumption on graphs resulting in a new form of GNN called SAT – its applicability on assembly modeling needs to be investigated.

**How to handle ambiguous edges?**  As introduced in Section 1, extracting a graph structure from an assembly model is an ambiguous task as only some mating relations may be given in the CAD system – some of them even serving other purposes than denoting meaningful connections (e.g., to simply support the designer's workflow). Consequently, extracting a graph structure from an assembly is not straightforward and when based on geometric proximity of parts an computational expensive approach. Even in a perfect world, where all parts are connected by mates in a meaningful way, this graph structure may be insufficient for the learning task (as mentioned in [23] and [24]) because parts that are far away from each other according the graph structure may have a certain relationship which is relevant for recommending next parts. *Graph rewiring* may be a promising solution for this issue, transforming the initial graph structure by adding and removing edges to improve information processing. Moreover, due to the novelty of the application domain, it is unclear whether a graph structure is really helpful for solving assembly modeling tasks, possibly a set-based approach could perform as well – this can also be systematically investigated using graph machine learning.

**How to improve intuitive sequence generation and interactive inference?**  Especially for part recommendation, the proposed sequence of part insertions needs to be intuitive in the eye of the designers that interact with the assembly modeling tool. There is not an obvious way to extract a sequence from a data set of graphs – as is done in generative graph models such as GraphRNN or GRAN. Either *all* possible insertion sequences (that leave the assembly connected) or a sample thereof need to be considered – as done in [6] – or the data sets need to be augmented with insertion sequences.

Finally, we encourage readers to investigate the data set [4] and identify similarities with their preferred data sets or the applicability of their methods that can address the above challenges. It contains real-world assemblies based on three disjoint part catalogs, represented as pseudonymized graphs (cf. Figure 1), as well as prepared samples for part recommendation consisting of partial assemblies and next needed parts. The complete designs can serve both for anomaly detection and part taxonomy creation. Table 1 presents a comparison of the catalog data.

**Table 1:** Key facts of the data sets based on the three part catalogs. For each metric column, the front part indicates the range of values and the back part the corresponding average. [6]

| catalog | #designs | #parts | node degrees | #nodes | #edges | graph diameter |
|---------|----------|--------|--------------|--------|--------|----------------|
| A | 11,826 | 1,930 | 1 - 9; $\emptyset$ 1.7 | 4 - 33; $\emptyset$ 6.1 | 3 - 32; $\emptyset$ 5.1 | 2 - 32; $\emptyset$ 4.45 |
| B | 11,895 | 3,099 | 1 - 13; $\emptyset$ 1.9 | 4 - 69; $\emptyset$ 18.2 | 3 - 68; $\emptyset$ 17.2 | 2 - 38; $\emptyset$ 10.06 |
| C | 11,943 | 1,924 | 1 - 16; $\emptyset$ 1.7 | 4 - 20; $\emptyset$ 6.7 | 3 - 19; $\emptyset$ 5.7 | 2 - 6; $\emptyset$ 2.94 |

## Author Contributions

**Carola Lenzen:** Conceptualization, Data curation, Methodology, Writing – original draft, Writing – review & editing

**Alexander Schiendorfer:** Conceptualization, Writing – original draft, Writing – review & editing

**Wolfgang Reif:** Funding acquisition, Supervision

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
