# OpenReview forum: "Graph Machine Learning for Assembly Modeling"
_logconference.io/LOG/2022/Conference — LoG 2022 Poster_

### Official Review · Reviewer_Gyda · 2022-10-14

**Overall Score:** 5
**Confidence:** 4

**Review:**

Summary: This extended abstract represents assembly modeling via graphs. The nodes represent the components. The edges are treated as temporal in nature and are associated with the development of an assembly. The author(s) provide a list of relevant tasks in assembly modeling, mention multiple limitations of the existing models, and provide solutions using graph neural networks and autoregressive models.

Strong points:

1. The idea of representing assemblies as graphs is interesting.
2. The mathematical formulations seem to be correct.
3. A proper list of potential applications has been provided.
4. 'Related Work' section is coherent and informative.

Weak points:

1. The writing and organization of the extended abstract could be improved.
2. A dataset is shared in Reference 3: Carola Gajek. ECML22 GRAPE Data. 7 2022. doi: 10.6084/m9.figshare.20239767.v1. URL https://figshare.com/articles/dataset/ECML22_GRAPE_Data/20239767. 1, 4. This might hamper the anonymity of this submission.
3. The inductive scenario needs more attention - where we deal with a new component, unseen in the training assemblies.
4. Lacks a performance metric to compare the proposed method with other state-of-the-art models.

Recommendation: Weak Rejection.

Supporting argument for recommendation:

Although this work discusses a potential application of graph machine learning and network science in assembly modeling, there are multiple questions that should be answered. An end-to-end example should be provided to fully explain the proposed method. The inductive scenario should be addressed in more detail - where an unseen component is dealt with. The solutions to the use cases like component recommendation and anomaly detection should be elaborated and formulated. Finally, a comparison with the existing approaches in terms of a well-defined performance metric should be added (I suggest adding a table or a figure).

Questions to the authors:

1. How do the node attributes for the components look like? Are they taken into consideration for the downstream tasks?
2. In the subsection 'How to handle ambiguous edges?', 'Graph rewiring' is mentioned. What kind of rewiring is being referred to here? Degree-preserved randomization?

Suggested improvements:

1. I suggest taking an example assembly and showing step-by-step how the proposed use cases are dealt with (Component Recommendation,  Anomaly Detection of Mismatching Components, etc.).
2. Propose a performance metric and compare the proposed method with computer vision-based methods and other state-of-the-art models.
3. Add citations for the embedding techniques which circumvent the information related to homophily.
4. Add citations for variational autoencoders or generative adversarial networks.
5. My understanding is that this work deals with graph generation processes like preferential attachment or random graphs. Study the basic network properties of the assembly graphs - degree distributions, clustering, etc. This will provide more insight into the graph properties of such assemblies and help in developing an interpretable model for assembly modeling.

Type of the paper: Extended abstract (4 pages)

---

### Official Review · Reviewer_EJjY · 2022-10-21

**Overall Score:** 8
**Confidence:** 3

**Review:**

**Summary**

This extended abstract provides an overview of the potential applications of graph ML for assembly modeling, an interesting summary of related work, and some open questions.

**Strength**

The authors provide great motivations for applying graph ML to assembly modeling and propose a couple of very interesting questions that could benefit from the involvement of the community. For instance, assembly modeling has many similar characteristics to molecule generation, and challenges in "ambiguous" edges can be interesting questions for "outlier" detection or uncertain measurement. Last but not least, I also appreciate the authors' attempt to bring the community's attention to a new assembly modeling dataset.

**Weakness**

While a dataset is introduced, it would also be helpful for the authors to showcase actual graph ML applications on the dataset. More importantly, designing a set of meaningful metrics for benchmarking could be more impactful.

**Recommendation**

This topic can be interesting for the graph ML community, so I give a clear acceptance.

---

### Official Review · Reviewer_sNrZ · 2022-10-21

**Overall Score:** 8
**Confidence:** 4

**Review:**

Summary. This extended abstract proposes to use GraphML to assist CAD design in assembly modeling, such as suggesting components for engineers, identifying mismatched components, and taxonomy creation. It formulates the process as a graph learning problem in general, where nodes are components and edges are potential components connected to existing components and discusses about major challenges and potential directions to solve them. The application domain is novel that deserves attention from the community and shows high values that can be helpful to improve CAD design efficiency, reduce manual costs, as well as optimize procurement of companies.

Recommendation. Given the novelty of the problem and plausible solutions discussed, I would recommend this extended abstract to be accepted in LoG 2022.

[Strengths]

S1. The extended abstract calls for attention to a novel application domain assembly modeling in CAD design, which has high potentials and values.

S2. The problem is clearly stated in graph learning and potential use cases and major challenge are well discussed.

S3. The whole draft is well-written and organized with proper related work cited.

[Need to improve]

N1. Related work is mainly from graph learning domain and existing assembly modeling methods are not discussed. See D1.

N2. The motivation why GraphML is preferred for assembly modeling over non-graph learning or even non-ML methods is not clear. See D2.

N3. The dataset (reference [3]) is mentioned as a contribution without any background or detail. See D3.


[Details]

D1. Related work is reviewed under graph learning domain (e.g., GRAN, GraphRNN, and molecule graph generation). Readers may wonder about the current approaches used in CAD software for assembly modeling.

D2. As discussed in the draft, homophily assumption usually does not hold in component suggestions. Shouldn’t it be a reason not to use graph learning? Also, compared to other methods such as traditional frequent item set mining, learning-to-rank (e.g., link prediction), non-graph ML/DL, etc., what is the special advance of graph learning in this scenario?

D3. This extended abstract mentions a CAD component dataset as a contribution and encourages readers to further investigate. The open-sourced dataset should be surely interesting to related readers. However, there is no background description or specification about the data, which can be confusing. A summary or an example would be preferred to discuss data.

[Questions]

Q1. Can the authors provide the raw data as well to show what real components and co-occurrence of components? After checking a few samples following reference [3]. It is only numerical data that can be loaded as arrays of numbers. This can limit the values for potential researchers or CAD engineers who work on the same domain, because they cannot do any further reasoning or map back to specific components in real-world.

Q2. Have you also considered about the role of domain experts in the process? I believe this is very important rather than a purely learned GraphML model. Take ambiguous edges for instance in Open Question 2, if our model can be guided by experts (or even a knowledge graph) when they compose components, the side effect of ambiguous edges can be mitigated.

For rebuttal, please address D1 – D3 and Q1 – Q2.

---

### Official Review · Reviewer_769W · 2022-10-23

**Overall Score:** 5
**Confidence:** 5

**Review:**

The paper presents correspondence of assembly models as graphs with heterogeneous nodes representing components of varied types and homogeneous edges representing connections between components. The paper also presents some use cases of Graph ML for assembly modeling, namely component recommendation, anomaly detection and taxonomy construction. The paper points out some key challenges and how it relates to ongoing research in Graph ML for other explored problems.

Strength:
1. The paper presents novel use-cases of graphs in assembly modeling that could potentially help to overcome manufacturing challenges.

Weaknesses:
1. Although the abstract mentioned about initial results (line 10), the paper did not present any results.
2. The well motivated use cases seem to be more of a sequential nature. The applicability of graphs, although not technically wrong, seems far-fetched. This is, however, noted by the authors too.

Recommendation: Reject
The applicability of graph for assembly modeling use-cases -- at least the ones presented in the paper -- seems not so promising. Although the paper presented 3 use-cases, it seemed to focus primarily on the 1st one (i.e., component recommendation). The next component recommendation is a sequential problem and does not necessarily depends on the graph structure at each step. Moreover, the notion of locality, smoothness over the graph structure/manifold is not well defined for this problem. As pointed out in the paper, the graph can either capture geometric proximity/'mates'/frequently co-occurring connections. There can be other potential use-cases of graphs in this domain which needs to be considered in future. But the submitted work as such do not provide helpful insights/results.

1. It is not clear how can the pointed-out challenges in Section 1 are addressed by graph ML. Please elaborate.
2. Line 34: The concept of "mates" is unclear even with the provided example.
3. Why isn't the sequence of components important in an assembly? The graph as modeled in section 1 will have no sequence information.
4. Line 44: What's $\mathcal{T}$ here?
5. The graph model seems to broad and ignoring valuable information such as connection type, geometry over edges.
6. The practicality of the component recommendation use case in Section 2.1 is unclear. The next component type recommendation use case seems more relevant and useful, however, what's the use of recommending component types for an existing set of components?

---

### Meta-Review · Area_Chair_Y1vn · 2022-11-09

**Confidence:** 4
**Recommendation:** Accept

**Meta Review:**

While the reviewers have a mixed response to this extended abstract, it is clear that a novel task for machine learning with graphs. The major weakness is that the abstract does not outline a clear reason why graphs are a particularly good fit for the problem compared to other machine learning methods (and thus of relevance to the LoG community).

For the reasons above, and given the nature of this submission as an extended abstract, I recommend acceptance.

I would like to encourage the authors to incorporate the feedback from the reviewers, and give emphasis in showing the relevance of this task to the LoG community.

---

### Decision · Program_Chairs · 2022-11-23

Accept (Poster)